# AN EVALUATION OF QUALITY AND ROBUSTNESS OF SMOOTHED EXPLANATIONS

## ABSTRACT

Explanation methods play a crucial role in helping to understand the decisions of deep neural networks (DNNs) to develop trust that is critical for the adoption of predictive models. However, explanation methods are easily manipulated through visually imperceptible perturbations that generate misleading explanations. The geometry of the decision surface of the DNNs has been identified as the main cause of this phenomenon and several *smoothing* approaches have been proposed to build more robust explanations. In this work, we provide a thorough evaluation of the quality and robustness of the explanations derived by smoothing approaches. Their different properties are evaluated with extensive experiments, which reveal the settings where the smoothed explanations are better, and also worse than the explanations derived by the common Gradient method. By making the connection with the literature on adversarial attacks, we further show that such smoothed explanations are robust primarily against additive $\ell_p$-norm attacks. However, a combination of additive and non-additive attacks can still manipulate these explanations, which reveals shortcomings in their robustness properties.

## 1 INTRODUCTION

Explanation methods attribute a numerical value to each data feature in order to quantify its relative importance towards the model's prediction. Such attributions help to better understand and trust complex models like deep neural networks (DNNs). In safety-critical tasks, such an understanding is a prerequisite to the deployment of DNNs, because a domain expert will never make important decisions based on a model's prediction unless that model is trustworthy. Moreover, explanations can help to understand the reasons behind the decision of a model, and when it comes to model debugging, they can reveal the presence of any spurious data correlations that may lead to faulty predictions during inference (Ribeiro et al., 2016).

In the context of image classification with deep neural networks, several explanation methods have been proposed based on the gradient with respect to input, also called gradient-based explanations (Baehrens et al., 2010; Bach et al., 2015; Selvaraju et al., 2017; Sundararajan et al., 2017; Springenberg et al., 2015). The explanation generated by these methods, *a saliency map*, highlights the parts of the image that contributed to the prediction. Recent work has shown that gradient-based explanations of neural networks can be fragile and can be easily manipulated via adversarially perturbed inputs (Ghorbani et al., 2019; Dombrowski et al., 2019; Heo et al., 2019; Viering et al., 2019; Kindermans et al., 2019). That is, one can find a small-norm perturbation to be added to an input ( often imperceptible), such that the focus of the explanation changes towards irrelevant features while the model's output remains unchanged. This, in turn, can make explanations inappropriate to help end-users gain trust in a model's prediction.

The large curvature of the decision surface of neural networks has been identified as one of the causes of fragility for gradient-based explanations (Ghorbani et al., 2019; Dombrowski et al., 2019; Wang et al., 2020). To make explanations more robust, a class of approaches proposed smoothing the explanation or making the decision surface of neural networks more smooth (Wang et al., 2020; Dombrowski et al., 2019; Ivankay et al., 2020). We refer to these approaches as *smoothing approaches*. It is worth mentioning that similar methods have been proposed in the context of adversarial robustness, with the aim of flattening the decision surface of neural networks in order to reach more robust predictions (Moosavi-Dezfooli et al., 2019; Qin et al., 2019).

Here, we provide a thorough investigation of the explanations derived by smoothing approaches in terms of *explanation quality* and *robustness*. We employ various tests to assess the quality of these explanations. Each test evaluates a desirable property for explanations, such as: sensitivity to changes in the model, fidelity to the predictor function, etc. In terms of robustness, we show that explanations derived by smoothing approaches only provide robustness against additive $\ell_p$ norm attacks. Specifically, in this work, we show that compared to additive attacks, attacks based on the combination of spatial transformation (Xiao et al., 2018) and/or color transformation (Laidlaw & Feizi, 2019) together with additive perturbations are more effective in manipulating these explanations. Our contributions can be summarized as follows:

- We study the effectiveness of smoothing approaches to achieve robust explanations. We present results on evaluating both the quality and robustness properties of smoothed explanations.

- We assess the quality of smoothed explanations via presenting the results of various quality tests. Our results demonstrate the pros and cons of smoothed explanations with respect to the following quality aspects: sensitivity to model parameters, class discriminativeness, Infidelity, and sparseness.

- We present results for different combination of additive and non-additive attacks, and show that they are able to manipulate explanations derived by smoothing approaches more successfully. Combining different types of perturbations to achieve stronger attacks has been a topic of investigation in the context of adversarial examples (Jordan et al., 2019). To the best of our knowledge, this is the first time such attacks have been used in the context of explanations.

**Related works.** There have been several works aiming to make explanations more robust. These works mostly focused on either modifying the explanation method itself or modifying the predictor model to achieve robust explanations. Wang et al. (2020) introduced Uniform Gradient, which is similar to Smooth Gradient unless it uses Uniform noise, and showed that it can hardly be manipulated by additive attacks. Dombrowski et al. (2019) proved that a network with soft-plus activations has a more robust Gradient explanation compared to a ReLU network, given that the parameter $\beta$ of the soft-plus function is chosen to be sufficiently small. Consequently, they proposed the $\beta$-smoothing approach in which they substitute the ReLU activations of a trained network by soft-plus functions with a small $\beta$ parameter. Wang et al. (2020) introduced a regularization term called *Smooth Surface Regularization (SSR)* to the training objective of a DNN. This training objective penalizes the large curvature of a DNN by regularizing the eigenvalue of the input hessian with the maximum absolute value. Moreover, they showed that adversarial training (Madry et al., 2018) also leads to more robust explanations. This fact can also be deduced from the results of (Moosavi-Dezfooli et al., 2019) as they showed that adversarial training leads to a significant decrease in the curvature of the loss surface with respect to inputs. Anders et al. (2020) proposed an attack in which they adversarially manipulate the model instead of the input in order to manipulate the explanation. Then they propose a modification to the existing explanation methods to make them more robust against such manipulated models. Lakkaraju et al. (2020) proposed a framework for generating robust and stable black box explanations based on adversarial training. Chen et al. (2019) introduced a regularization term to the training objective of neural networks to achieve robust Integrated Gradient explanations. Finally, Dombrowski et al. (2020) developed a theoretical framework to derive bounds on the maximum manipubality of explanations and proposed three different techniques to boost the robustness of explanations. In this work, we show that the robustness of smoothed explanations can be affected by employing a combination of additive and non-additive attacks. Furthermore, we present a through evaluation of the different quality aspects of smoothed explanations.

## 2 BACKGROUND

First, we provide the definition of an explanation map and then briefly describe the explanation methods we used in this paper. Then we continue with introducing the attacks to explanations and the smoothing approaches we are going to study in this paper.

Consider a model $f : \mathbb{R}^d \to \mathbb{R}^K$ which classifies an input $\mathbf{x} \in \mathbb{R}^d$ into one of the $K$ classes. *An explanation map*, denoted by $h_f(\mathbf{x}) : \mathbb{R}^d \to \mathbb{R}^d$, associates a score to each feature of the input

indicating the relevance of that feature towards the model's prediction. For instance, in the context of image classification, saliency maps associate a score to each pixel of the input image resulting in a heatmap that highlights important regions of the image leading to the model prediction. In this work, we focus on the gradient-based explanations and mainly on the Gradient method. Given a model $f$ and an input $\mathbf{x}$, the Gradient explanation is defined as $\nabla_{\mathbf{x}} f(\mathbf{x})$. Since other gradient-based explanation methods make use of the gradients with respect to input, we argue that our results could be extended to those explanation methods as well. We will also consider two smoothed variants, namely Smooth (Smilkov et al., 2017) and Uniform Gradient (Wang et al., 2020) methods.

## 2.1 ATTACKS TO MANIPULATE EXPLANATIONS

Similarly to common adversarial attacks (Goodfellow et al., 2015; Moosavi-Dezfooli et al., 2016; Szegedy et al., 2014), recent work has shown that explanations can also be manipulated by adding a small and almost imperceptible perturbation to the input (Ghorbani et al., 2019; Dombrowski et al., 2019). We refer to this class of attacks as *explanation attacks*. There have been various formulations for explanation attacks (Ghorbani et al., 2019; Dombrowski et al., 2019). In this work, we will use the formulation introduced by Dombrowski et al. (2019). In this attack, the attacker tries to find a perturbed input for which the explanation is manipulated to be very similar to a given target explanation map while the output of the model remains approximately unchanged. Note that the target map could be any heatmap in general; however, we used the explanation of a target image as a target map in this work. Below, we will give a formal definition of this attack.

**Definition 1** (Targeted manipulation attack). *An explanation $h_f(\mathbf{x})$ for model $f(\mathbf{x})$ is vulnerable to attack at input $\mathbf{x}$ if there exist a perturbed input $\mathbf{x}_{adv}$, such that $h_f(\mathbf{x}_{adv})$ is similar to a given target map $h^t$ but the model's output remains unchanged. An attacker finds $\mathbf{x}_{adv}$ by minimizing the following objective function:*

$$\mathcal{L} = \left\| h_f(\mathbf{x}_{adv}) - h^t \right\|^2 + \gamma_1 \left\| f(\mathbf{x}_{adv}) - f(\mathbf{x}) \right\|^2 + \gamma_2 \mathcal{L}_{reg}(\mathbf{x}, \mathbf{x}_{adv}) \tag{1}$$

The first term in (1) ensures the similarity of the manipulated explanation to the target map, the second term ensures the similarity between the model output for the original and perturbed inputs, and the third term regularizes the perturbation to ensure perceptual similarity between the original and perturbed images. Note that $\mathcal{L}_{reg}$ is defined by the attacker according to the type of the perturbation. The relative weighting of the terms in (1) is controlled by the hyper-parameters $\gamma_1$ and $\gamma_2$.

## 2.2 TOWARDS ROBUST EXPLANATIONS

Recent works have tried to define the robustness of explanations in terms of the sensitivity of input gradients to changes in the input data (Wang et al., 2020; Dombrowski et al., 2019). Wang et al. (2020) define the robustness of explanations by the Lipschitz continuity coefficient of the input gradients; a smaller coefficient means that the explanation is less sensitive to the changes in the input and hence more robust. In this regard, a class of approaches to generate robust explanations have been proposed in the recent works, which are either based on smoothing out the explanation maps or flattening the decision boundary of the model itself. Broadly, these approaches can be classified into two categories: (1) *Post-hoc* approaches do not require retraining of the network and can be applied as a post-processing step. (2) *Ad-hoc* approaches to robust explanations require retraining of the network and hence are more costly.

In this work, we consider Smooth Gradient (Smilkov et al., 2017), Uniform Gradient (Wang et al., 2020), and $\beta$-smoothing (Dombrowski et al., 2019) as post-hoc approaches. The first two methods involve smoothing the explanation map, while the third one smooths the decision surface of the model. All three approaches act on pre-trained models, and hence are characterized as post-hoc. Among the ad-hoc methods, we study the explanations generated by adversarially trained networks, and networks trained with curvature regularization (CURE) (Moosavi-Dezfooli et al., 2019), which is a similar approach to SSR (Wang et al., 2020)[1].

---

[1]We experiment only with CURE, because with the publicly available code of SSR we were not able to reproduce the results in (Wang et al., 2020).

# 3 EVALUATING POST-HOC APPROACHES

Here, we begin by evaluating the *quality* of explanations derived by post-hoc approaches that do not require retraining of the network. Then, we evaluate the *robustness* of these explanations by presenting results on effective non-additive attacks to manipulate them. For all of the experiments in this section, we used a VGG-16 network trained on ImageNet (Russakovsky et al., 2015), and for generating the explanation maps we used the Captum (Kokhlikyan et al., 2020) package. Moreover, for the $\beta$-smoothing approach we always set $\beta = 0.8$ as suggested in (Dombrowski et al., 2019).

## 3.1 QUALITY OF EXPLANATIONS OF POST-HOC APPROACHES

To evaluate and compare the quality of the explanations, we use various quality tests presented in the literature. In general, assessing the quality of an explanation is a challenging task and each quality test only evaluates a specific quality aspect of an explanation. Therefore the assemblage of quality tests helps to understand which quality aspects of the explanations are improved and which are deteriorated by the smoothing approaches.

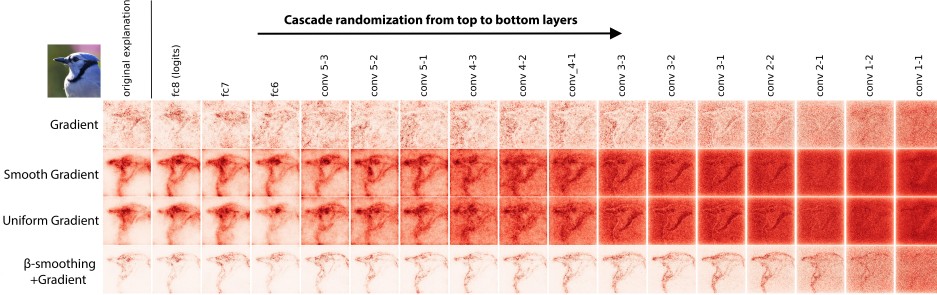

Figure 1: Cascade randomization of the VGG-16 (ImageNet) layers. The first column shows the original explanation for the image "Jay" bird. Each subsequent column shows the effect of randomization of the parameters of the network up to that layer (inclusive) on the explanations.

**Cascade randomization of model parameters.** Adebayo et al. (2018) argued that it is desired for an explanation to be sensitive to the changes in the model parameters. They proposed a model parameter randomization test to assess this sensitivity. In this test, the parameters of a model are progressively randomized from the top layer (logits) to the bottom layers. In each step of randomization, the explanation from the resulting model is compared against the explanation from the original model. Randomizing the model parameters means losing what the model has learned from the data during training. Therefore, we expect a "good" explanation to be destroyed in this process. However, if an explanation is insensitive to the randomization of the model parameters, then it is not deemed appropriate for debugging the model under erroneous predictions.

The visual results of this test for Gradient explanation and post-hoc approaches are shown in Figure 1. More examples of this test can be found in the Appendix. One can observe that the explanations derived from post-hoc approaches show less sensitivity to the randomization of model parameters than compared to the Gradient method. This can also be verified by the Spearman rank correlation between the original and randomized explanations shown in Figure 2. We observe that for the smoothed explanation methods, the original and randomized explanations have a high rank correlation after the randomization of the top layers of the network. *These results highlight that using Smooth Gradient, Uniform Gradient, and $\beta$-smoothing to achieve a more robust explanation can come at the expense of having explanations that are less sensitive to model parameters.*

**Class sensitivity of explanations.** A good visual explanation should be able to localize the image regions relevant to the target category, i.e., it should be *class discriminative* (Selvaraju et al., 2017). This is particularly significant when dealing with images containing more than one object. To assess the class discriminativeness of an explanation we used a quality test equivalent to the pointing game (Zhang et al., 2016). We sampled images from the MS COCO dataset (Lin et al., 2014), containing two objects that are also present among the ImageNet class labels. For this test we only keep

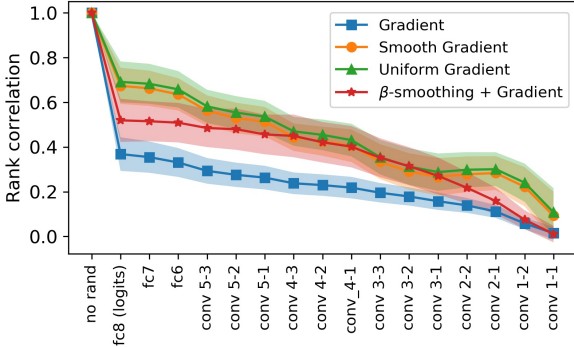

Figure 2: Spearman rank correlation between the original and randomized explanation derived for randomization up to the layer indicated by the x-axis. A higher rank correlation value indicates a higher similarity, i.e, *the higher the curve of an explanation method, the less sensitive it is to model parameters.* The results are averaged over 1000 Images from Imagenet and the shaded area around each curve indicates the standard deviation.

Table 1: Ratio of the explanation top-20 values included in the segmentation mask of each object. Note that the columns "obj 1" and "obj 2" are for the explanations computed for the top predicted class and the class corresponding to the second object in the image respectively. The results are averaged over 60 samples.

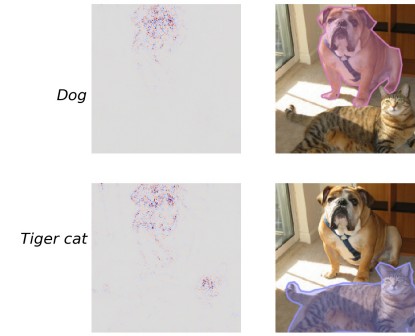

Figure 3: An example of $\beta$-smoothing explanation generated for the target category "dog" and "cat" (left), and the segmentation mask of each object from the COCO dataset (right).

| Explanation Method | obj 1 | obj 2 |
|---|---|---|
| Gradient | 0.6 | **0.49** |
| Smooth Gradient | 0.54 | 0.37 |
| Uniform Gradient | 0.57 | 0.38 |
| $\beta$-smoothing + Gradient | **0.62** | 0.45 |

the samples for which one of the objects in the image is the top predicted class by the network and the other object is among the top 20 predicted classes by the network. We compute the explanation maps for each of the class labels corresponding to the objects. Using the segmentation mask of the objects provided in the dataset as ground truth, we compute what percentage of the top-20 values in the explanation maps generated for each target category are inside the corresponding segmentation masks. The results of this test are shown in table 1 and a visual depiction of this test is given in Figure 3. These results indicate that the smoothed explanation methods are less discriminatory when generated for the target class label that has a lower probability. *This suggests that in terms of class discriminativeness of explanations, the post-hoc smoothing approaches investigated in this paper are inferior to the Gradient method.*

**Sparseness of explanations.** To create explanations that are human-accessible, it is advantageous to have a *sparse* explanation map (Molnar, 2019), i.e, only the features that are truly predictive of the model output should have significant contributions, and irrelevant features should have negligible contributions. Sparse explanations are more concise because they only include features with significant contribution making it simpler for end-users to understand the reasons for a specific prediction of the model (Chalasani et al., 2020). To measure the sparseness of an explanation map, we applied the Gini Index on the absolute value of the flattened explanation maps. The Gini Index is a metric that measures the sparseness of a vector with non-negative values (Hurley & Rickard, 2009). By definition, the Gini Index take values in $[0, 1]$ with higher values indicating more sparseness. Table 2 shows the average Gini Index of the Gradient, Smooth Gradient, Uniform Gradient, and

Table 2: Average Gini Index for the explanations of a VGG-16 network (averaged over 1000 samples). A **larger** Gini Index indicates more sparseness and hence a more concise explanation.

|  | Gradient | Smooth Gradient | Uniform Gradient | $\beta$-smoothing |
|---|---|---|---|---|
| Gini Index | $0.56 \pm 0.038$ | $0.34 \pm 0.053$ | $0.35 \pm 0.058$ | $\mathbf{0.65} \pm 0.067$ |

Table 3: Average Infidelity for the explanations of a VGG-16 network (averaged over 100 samples). A **lower** Infidelity value indicates better fidelity of the explanation to the predictor function.

|  | Gradient | Smooth Gradient | Uniform Gradient | $\beta$-smoothing |
|---|---|---|---|---|
| Infidelity | $1.43 \pm 1.52$ | $1.42 \pm 1.52$ | $1.42 \pm 1.52$ | $\mathbf{1.00} \pm 0.88$ |

$\beta$-smoothing computed for 1000 randomly sampled images from ImageNet. *The results show that compared to the Gradient method, Smooth Gradient and Uniform Gradient provide less concise explanations, whereas $\beta$-smoothing actually improves the sparseness of the explanations as compared to the Gradient method.*

**Explanation Infidelity.** Introduced in Yeh et al. (2019), this metric captures how the predictor function changes in response to significant perturbations to the input and is defined as the expected difference between the two terms: 1) the dot product of the input perturbation and the explanation and 2) the difference between function values after significant perturbations to the input. The metric generalizes the completeness axiom (Shrikumar et al., 2017; Sundararajan et al., 2017) because it allows for different types of perturbations which could be of interest depending on the problem and the dataset. We use the infidelity metric to compare the effect of post-hoc smoothing approaches on the fidelity of explanations to the predictor function. As suggested in (Yeh et al., 2019), we used the square removal perturbation to compute the infidelity of explanations for randomly selected images from ImageNet. Table 3 shows the results for the post-hoc approaches. A lower infidelity value indicates better fidelity of the explanation to the predictor function. *The results suggest that the degree of smoothing used to robustify explanations, also improves their infidelity.* Therefore with respect to the Infidelity metric, all of the smoothed explanations investigated in this section are superior to the Gradient method. This finding is also in line with the results of Yeh et al. (2019), i.e., that modest smoothing improves the infidelity of explanations.

## 3.2 ROBUSTNESS OF EXPLANATIONS OF POST-HOC APPROACHES

Now, we will evaluate the robustness of Smooth Gradient, Uniform Gradient, and $\beta$-smoothing explanations. We present attacks composed of additive and non-additive perturbations, and show that they are more effective than additive attacks to manipulate explanations. The non-additive attacks we employed are spatial transformation attacks (Xiao et al., 2018), and recoloring attacks (Laidlaw & Feizi, 2019). See the Appendix B for a brief description of each of these attacks. In the rest of this paper, we refer to the additive attack as *Delta*, spatial transformation attack as *StAdv*, and recoloring attack as *Recolor*.

We used the projected gradient descent (PGD) algorithm to optimize the objective function (1)[2]. In our experiments, we evaluate three combinations of attacks, namely Delta, Delta+StAdv, and Delta+StAdv+Recolor, against the explanation of a VGG-16 network trained on ImageNet (Russakovsky et al., 2015). See Appendix C.1 for the details about the $\ell_\infty$ norm for each type of the perturbations and the hyper-parameters used in each attack setting.

We use two metrics to evaluate the attacks: (1) The Cosine Distance metric (cosd) to evaluate the similarity between the target and manipulated explanations (Wang et al., 2020). A lower cosine distance corresponds to a lower $\ell_2$ distance between the target and manipulated explanations indicating a higher similarity. The range of the values for cosd is between 0 and 1. (2) The LPIPS metric for

---

[2]As discussed in (Ghorbani et al., 2019; Dombrowski et al., 2019), to avoid zero-valued gradients when optimizing (1), we have to replace the ReLU activation with its smooth approximation. In this work, we used a soft-plus function with $\beta = 100$.

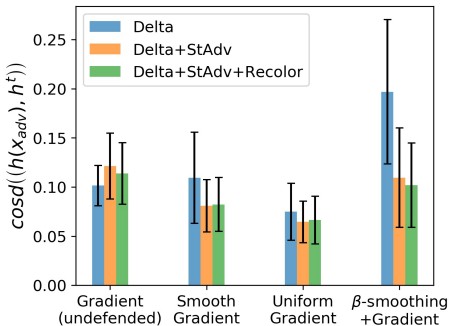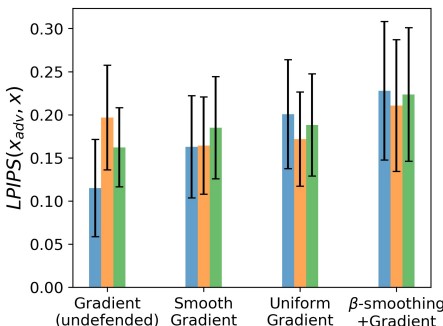

Figure 4: Evaluation with cosd (left) and LPIPS metrics for attacks against post-hoc approaches. For both cosd and LPIPS, smaller values indicate higher similarity. All results are averaged over 200 images from ImageNet. The black bars indicate standard deviation.

quantifying the *perceptual* similarity between images (Zhang et al., 2018). A lower LPIPS value indicates higher similarity.

Figure 4 shows the cosine distance between the target and manipulated explanations, and the perceptual similarity (LPIPS) between the perturbed and original images for each attack setting. We can observe that Delta+StAdv, and Delta+StAdv+Recolor attacks are more effective than Delta attacks to manipulate $\beta$-smoothing explanations, i.e, with a less perceptible perturbation (lower LPIPS value), we can reach a cosd value between manipulated and target explanations very close to the cosd value when attacking the Gradient method. The effect of the non-additive attacks is less significant on the Smooth and Uniform Gradient methods, however we can still observe improvements in the cosd values under these attacks. Taken together, these results show that *Smooth Gradient, Uniform Gradient, and $\beta$-smoothing explanations are more vulnurale to non-additive attacks and hence such attacks should be considered as a threat to the robustness of these methods*. As an example, we can visually see the effectiveness of Delta+StAdv+Recolor attack against differnt explanation methods in Figure 5.

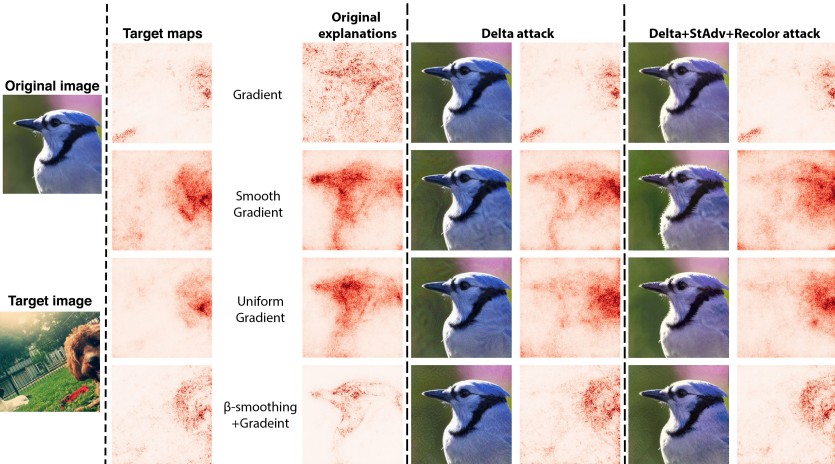

Figure 5: An example of visual comparisons of explanation attacks against different methods. Under each attack setting, the left column shows the perturbed image and the right column shows the corresponding perturbed explanation.

## 4    EVALUATING AD-HOC APPROACHES

Here, we recreate the experiments of Section 3 for the ad-hoc approaches. We study the explanations of networks trained with curvature regularization (CURE) (Moosavi-Dezfooli et al., 2019),

and adversarial training (Madry et al., 2018). Training with CURE, regularizes the eigenvalue of the input hessian with maximum absolute value and is similar to SSR, which was shown to improve the robustness of explanations against additive attacks (Wang et al., 2020). Adversarial training also smooths the decision surface and can provide more robust explanations.

For the experiments in this section, we used a ResNet-18 network trained with CURE and an adversarially trained ResNet-18 network trained on adversarial examples with $\ell_\infty$ norm of the perturbations upper bounded by $8/255$ (Engstrom et al., 2019). Both networks are trained on CIFAR-10 dataset (Krizhevsky, 2012).

### 4.1 QUALITY OF EXPLANATIONS OF AD-HOC APPROACHES

**Cascade randomization of model parameters.**   We evaluate the sensitivity of explanations of the networks trained with CURE and adversarial training using the cascade randomization of model parameters test.

The Spearman rank correlation between the original and randomized explanations is shown in Figure 6. These Results show the explanation of an adversarially trained network is less sensitive to model parameters. *This suggests that the explanation of an adversarially trained network cannot be helpful to debug a model when it is making a wrong prediction.*

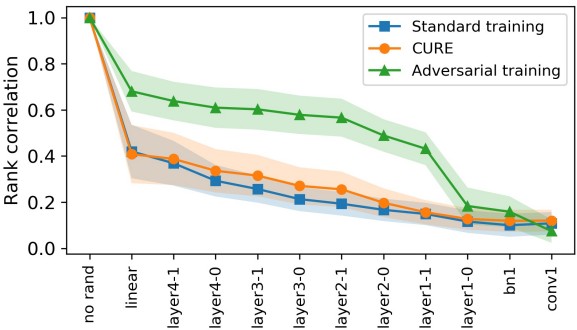

Figure 6: Spearman rank correlation between the original and the randomized explanations (averaged over 1000 Images from CIFAR-10). A higher rank correlation value indicates a higher similarity, i.e, *the higher the curve of an explanation method, the less sensitive it is to model parameters.*

**Sparseness of explanations.**   We compare the sparseness of the explanations derived by ad-hoc approaches, using the Gini Index metric. Table 4 compares the Gini Index for the explanations of networks trained with different training objectives. These results show that adversarial training helps to improve the sparseness of explanations as compared to standard training. Hence the explanations of an adversarially trained network are more *concise*. This is in line with the results of Chalasani et al. (2020) as well. However, the rsults of Table 4 indicates that training a network with CURE does not help to improve the sparseness of explanations as compared to standard training.

Table 4: Gini Index for the explanations of a ResNet-18 network (averaged over 1000 images from CIFAR-10). A **larger** Gini Index suggests a more concise explanation.

|  | Standard training | CURE | Adversarial training |
|---|---|---|---|
| Gini Index | $0.54 \pm 0.035$ | $0.54 \pm 0.045$ | $\mathbf{0.71} \pm 0.054$ |

**Explanation Infidelity.**   To compare the fidelity of explanations derived by ad-hoc approaches to the predictor function, we used the Infidelity metric with square perturbation (Yeh et al., 2019). Table 5 shows the results for randomly selected images from CIFAR-10. A lower infidelity value indicates better fidelity of the explanation to the predictor function. From these results, we can observe that training a network with CURE and adversarial training helps to improve the explanation Infidelity. Therefore with respect to the Infidelity metric, the ad-hoc smoothing approaches investigated in this section improve the explanation Infidelity as compared to standard training.

Table 5: Infidelity of the explanations of a ResNet-18 network (averaged over 1000 images from CIFAR-10). A **lower** Infidelity value is better.

|  | Standard training | CURE | Adversarial training |
|---|---|---|---|
| Infidelity | $5.69 \pm 4.44$ | $\mathbf{0.59} \pm 0.34$ | $1.56 \pm 0.76$ |

## 4.2 ROBUSTNESS OF EXPLANATIONS OF AD-HOC APPROACHES

Now, we evaluate the improvement of robustness via ad-hoc approaches. We present results for Delta, Delta+StAdv, and Delta+StAdv+Recolor attacks against explanations of the networks trained with CURE and adversarial training. Figure 7 shows the results of these attacks. For the adversarially trained network, we can observe that non-additive attacks can more effectively manipulate explanations compared to the additive attacks. However, even with the strongest attack setting we still cannot get close to the cosd value reached by attacking the explanation of the network trained in standard way. For the attacks against the explanation of the network trained with CURE, the effect of non-additive attacks are less significant in terms of the cosd value, however we can still observe that such attacks can reach similar cosd values with perceptually less visible perturbations.

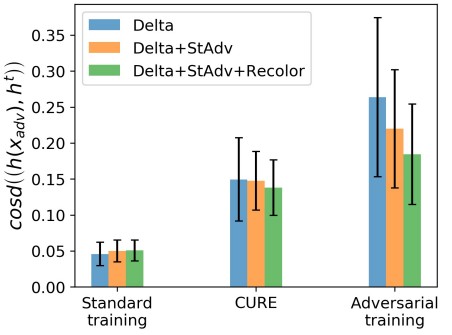 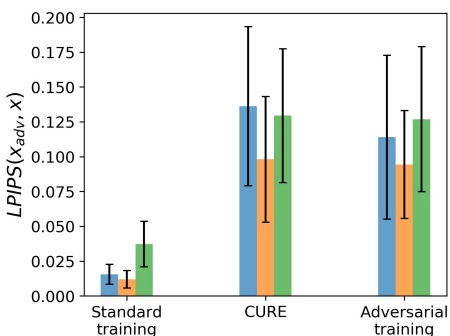

Figure 7: Evaluation with cosd (left) and LPIPS (right) metrics for the attacks against ad-hoc approaches. For both cosd and LPIPS, smaller values indicate higher similarity. All results are averaged over 1024 images from CIFAR-10. The back bars indicate standard deviation.

## 5 CONCLUSION

We have evaluated two aspects of smoothed explanations: a) explanation quality, and b) robustness of explanation. In terms of explanation quality, we performed a thorough evaluation of four quality aspects: sensitivity to model parameters, class discriminativeness, sparseness, and infidelity. Our results show that the smoothed explanations investigated in this paper perform worse than those of the Gradient method in terms of sensitivity to model parameters and class discriminativeness. On the other hand, we show that using such smoothing methods helps to improve explanation Infidelity and sparseness.

We further looked at the robustness of explanations, when inputs are perturbed by a combination of additive and non-additive attacks. To the best of our knowledge, this is the first time such attacks are used to manipulate explanations. Our experimental results highlighted the fact that non-additive attacks are still a threat for explanation methods, including the smoothed ones. These results also point us to the fact that many problems in explanation robustness can be addressed by making analogies with the area of prediction robustness. As these two areas are closely related, the solutions already explored in prediction robustness can be potentially helpful to study explanation robustness. This will be the focus of our future work.

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
