# OpenReview forum: "An evaluation of quality and robustness of smoothed explanations"
_ICLR.cc/2022/Conference — ICLR 2022 Submitted_

### Official Review · Reviewer_xHez · 2021-10-21

**Correctness:** 2
**Technical Novelty And Significance:** 3
**Empirical Novelty And Significance:** 2
**Recommendation:** 5
**Confidence:** 4

**Main Review:**

Novelty and Significance:
The novelty and significance of this paper are in the experimental evaluation of "smoothing explanation approaches" using diverse evaluation measures. The major novelty and significance of this paper can come from section 3.2 that aims to show non-robustness of the smoothing methods. However, the experimental results do not show a significant difference between the three attacks, which degrades the novelty and significance. Section 3.1. experiment and result are similar to those in prior works, but still can provide somewhat meaningful observations  (i.e. insensitivity) about the smoothing approaches if error bar is provided.

Technical soundness: The experimental evidence is not fully supporting the authors' claims. Additional experiments and in-depth discussion are needed.
- Table1-3, 5: Please provide error bar (i.e. empirical confidence interval). If error bar is not small enough to distinguish the difference, please use more samples for evaluation.
- Figure 4, 7. The two metrics are not consistent with each other. For example, in figure 4, the non-additive attack is shown as effective for Smooth gradient and $\beta$-smoothing in the left figure but not in the right figure. Why do those two metrics show different results?
- Figure 4, 7. Each smoothing approach shows no significant differences between the three attacks. It is quite hard to say that those methods are less robust to other types of attacks based on this experiment. Please provide more experimental evidence or use more samples to reduce the error bar.

Writing and Clarity: Minor corrections are needed.
- Page 1. The first sentence 'Explanation methods attribute a numerical value ... ' is not always true. There are other explanation methods not using attribution scores. Please modify this sentence.
- Page 2. Infidelity -> infidelity


**Summary Of The Paper:**

This paper evaluates the quality and robustness of explanations of three post-hoc smoothing approaches (Smooth Gradient, Uniform Gradient, B-smoothing), and two ad-hoc smoothing approaches (CURE, Adv). It evaluates the quality of explanations based on the model parameter sensitivity, class sensitivity, sparseness, infidelity. It also evaluates the robustness of explanations to combinations of additive, spatial transformation, and recoloring attacks by comparing similarities between target and explanation maps. All evaluation is performed on publicly available benchmark datasets such as ImageNet and MS COCO. Based on their experimental results, the authors made several claims about the quality and robustness compared to the vanilla gradient method. For example, the authors claim that most of the smoothing methods are less sensitive to perturbation of model parameters so they may not be helpful to debug a model, and they may not be "robust" to non-additive types of attacks.

**Summary Of The Review:**

The novelty and significance of this paper are in the experimental evaluation of robustness of the smoothing explanations that were known to be robust and any quality degradation trade-off. However, the experimental evidence is not fully supporting the authors' claims. The paper should provide experimental evidence showing significant robustness reduction. Also in-depth discussion about the experimental results is needed.

---

> ### Author Response · Authors · 2021-11-17
> **The non-additive attacks can make the $\beta$-smoothing explanations as non-robust as the gradient method.**
>
> We thank the reviewer for their feedback. Here are our comments on the three main points raised by the reviewer:
>
> - We have been augmenting the results as much as possible given the time we had and the computational power we had at hand. We have also added standard deviations to all of the results in the paper. All of the new results and changes are applied in the rebuttal draft which will be submitted shortly.
>
> - Note that in figures 4 and 7, the left figure shows the cosine distance value between the perturbed and the target explanation and the right figure shows the perceptual similarity between the perturbed and original image measured by the LPIPS metric. Therefore, the results should be interpreted as follows: For smooth gradient, uniform gradient, and $\beta$-smoothing, the non-additive attacks are more successful as they result in a smaller cosd value (i.e, the perturbed explanation is more similar to the target), and at the same time, compared to the additive attack, they do not result in more visually perceptible perturbation (the LPIPS value is almost at the same level as the additive attack or even smaller)
>
> - For the $\beta$-smoothing method, the new non-additive attacks can actually create 0.1 difference in the cosine distance value compared to the additive attack. By comparing the results of the attacks against the gradient and the $\beta$-smoothing methods (Figure 4 left), we can observe that using the non-additive attacks against the $\beta$-smoothing method, we can actually reduce the cosd value to the same level that was achieved by attacking the gradient method (which is not robust). In other words, attacking the $\beta$-smoothing method with the combination of additive and non-additive attacks, makes it as non-robust as the gradient method. We achieve this result while we are not losing the original and adversarial image similarity, compared to when attacking the $\beta$-smoothing method with the additive attack (Figure 4 right). For the case of the smooth and uniform gradient methods, the additive attack can already reduce their cosd value to the same level as the gradient method. Therefore, for these methods, the difference that the non-additive attacks can make is less significant but they still can reduce the cosd value as compared to the additive attack (Figure 4 left, second and third column).

---

> > ### Author Response · Authors · 2021-11-26
> > **Could you please share with us whether your concerns have been addressed or not?**
> >
> > Dear reviewer,
> >
> > Thanks again for your feedback and comments. In our reply to your previous comments, we tried to address your questions and concerns. We also updated the manuscript and increased the samples for all of the experiments as much as possible. We also provided error bars for all results. We would very much appreciate it if you could share with us your opinion about our reply. Did our reply address your questions and concerns?
> >
> > Thanks a lot.

---

### Official Review · Reviewer_Aokc · 2021-11-02

**Correctness:** 3
**Technical Novelty And Significance:** 3
**Empirical Novelty And Significance:** 3
**Recommendation:** 6
**Confidence:** 4

**Main Review:**

### Strengths
- a broad set of evaluation metrics is used to measure the quality
- hyper-parameters are well-documented, reproducibility is high
- methods from prior work are studied thoroughly for their application
- the paper is structured and written well


### Weaknesses
- the sample set of almost all experiments is very small and should be increased
    - currently for post-hoc approaches on ImageNet:
        - 128 samples for the similarity to the target map
        - 100 samples for cascade randomization
        - 20 (!) samples for the segmentation
        - 100 samples for sparseness with the GINI index
        - 100 samples for infidelity
    - currently for the ad-hoc approaches on CIFAR-10:
        - 320 samples for the similarity to the target map
        - 100 samples for cascade randomization
        - 100 samples for sparseness with the GINI index
        - 100 samples for infidelity

- errors should be reported for the segmentation experiment in table 1, the
  cascade randomization (Figure 2 & 6), the GINI index (Table 2 & 4) and the
  infidelity (Table 3 & 5)

- only a single model on a single dataset is used for each experiment, which should be increased
    - VGG16 on ImageNet for post-hoc approaches
    - ResNet-18 on CIFAR-10 for ad-hoc approaches

- only a single run of the experiments is conducted with only a single seed
  (model parameters)
    - to test for statistical significance, the experiments should, in addition
      to more models and a higher sample size, be conducted multiple times with
      multiple seeds

- are the error bars in Figures 4 and 7 percentiles or the standard deviation?
- the errors in Figures 4 and 7 are very large ("compatible results" most of
  the time), but this is not discussed

- in the segmentation experiment in Table 1 only the 20 pixels with highest
  attribution values are investigated
    - why do the authors not use all pixels' attribution values?
    - if there is any reason behind this, it should be discussed, otherwise all
      pixels should be used
    - this approach measures spikey attribution maps in a different way than
      smooth ones, since the spikey attribution maps will have more of their
      total attribution scores measured

- in Table 3 the infidelity for Smooth and Uniform are only marginally better
  (1.43 vs. 1.42), yet in the text this is simply described as an improvement
  (this marginal improvement will probably even become less significant with
  errors reported)

- it seems a little suspicious that Smooth and Uniform Gradient have the exact
  same values for GINI Index and infidelity in Tables 2 & 3, though this may
  just be by chance

- for the similarity in Figures 4 & 7, it may give more insights to also
  compute the distance to the original, ie. dist(h(x_{adv}, h(x)))

- why is the segmentation experiment is not conducted for the ad-hoc methods?

- minor: a label "higher is better" for figures 4 & 7 could improve readability


**Summary Of The Paper:**

The authors empirically evaluate the quality and robustness of 3 post-hoc and
2 ad-hoc approaches for robustification of gradient-based attributions under a
combination of 3 adversarial attack approaches which they use to target the
attributions.
They evaluate the robustness via cosine-distance and LPIPS to a target-heatmap,
and the quality via spearman correlation to the original under cascade
randomization of model parameters, the sparseness of the attribution maps using
the GINI index, and the explanation infidelity.

**Summary Of The Review:**

While the results of this work are potentially interesting, this paper has many
short-comings in its empirical evaluation, which is especially problematic
since it is the paper's main contribution. With small sample sizes, most errors
unreported, only a single model and dataset, and only a single run, there is a
lot of room to improve this work. As these changes involve running many more
experiments, for the sake of improving this work, I recommend a score of 3:
reject, not good enough.

---

> ### Author Response · Authors · 2021-11-17
> **We have augmented the draft with new results with more samples and added standard deviation values to all the results.**
>
> We thank the reviewer for their detailed feedback. Our comments on the weakness points raised by the reviewer are as follows:
>
> - Small sample set for the experiments and error bars: We have been augmenting the results as much as possible given the time we had and the computational power we had at hand. We have also added standard deviations to all of the results in the paper. All of the new results and changes are applied in the rebuttal draft which will be submitted shortly.
>
> - In the experiments, we used two network architectures, namely, VGG-16 and Resnet-18. These networks have a similar structure to other CNN and Resnet networks, so we believe that the results are representative and can be extended to other such architectures as well.
>
> -  We did not fix the seed for our experiments. We used different seeds for each run and the reported results are averaged over these runs.
>
> - we thank the reviewer for their detailed comment. Those error bars indicate standard deviation. We have made this clear in the rebuttal draft.
>
> - The results in Figures 4 and 7 show the average and standard deviation over the different images on which the experiment was run. Depending on the original and target explanation, the targeted manipulation attack could be hard to perform. This in turn can cause large differences between the final cosd values for each attack. As we increase the sample size these error bars will converge to a specific value but they do not necessarily converge to zero. We also saw this in our results as well, i.e, as we increased the sample size from the standard deviation values started to converge.
>
> - In this experiment, the goal was to evaluate the ability of the explanations to discriminate between different classes in an image. Therefore, we had to keep a subset of images from MS COCO dataset, for which the network was giving a high rank (or probability) for the class labels corresponding to the objects in the image. The goal is to observe when we compute the explanation for a specific target class, how well the explanation can detect the regions in the image related to that target class. To achieve this, we performed an experiment similar to the pointing game [1]. For the evaluation, we used the segmentation mask as ground truth and computed how many of the explanation top values land inside the segmentation mask. This gives us an indication of how the focus of the explanation aligns with the corresponding object’s segmentation box. Therefore, it would not have been interesting for us to use all of the attribution values since here we care only about the intersection of the explanation's top values with the mask.
>
> - We thank the reviewer for their detailed feedback. For the explanation of the images from Imagenet dataset, Smooth and Uniform Gradient only create marginal improvement in terms of infidelity. We have updated the text in the rebuttal draft accordingly. It is worth mentioning that the authors in [2] have also done a similar experiment on Imagnet dataset with the Smooth gradient method and their results also show that Smooth gradient only creates marginal improvement in terms of the infidelity metric.
>
>
> - In the explanation attacks we used in this work, the objective is to manipulate the explanation such that it gets as close to a target explanation map as possible. Therefore, for comparing these attacks against different methods, it is fairer to consider the distance to the target explanation which was included in the objective of the attack. Nevertheless, for the sake of completeness, we reported the distance to the original explanation for the attacks against ad-hoc methods in the appendix of the rebuttal draft.
>
> - For the ad-hoc methods we considered adversarial training and curvature regularization. These methods cannot yet reach state-of-the-art results on the Imagenet dataset, and have mainly shown to be effective for smaller datasets like CiFAR-10. Therefore, we couldn’t perform the segmentation experiment for these methods, as there is no segmentation dataset available for small images in the size of CIFAR-10 images.
>
> [1] Jianming Zhang, Zhe L. Lin, Jonathan Brandt, Xiaohui Shen, and Stan Sclaroff. Top-down neural attention by excitation backprop - ECCV 2016
>
> [2] Chih-Kuan Yeh, Cheng-Yu Hsieh, Arun Sai Suggala, David I. Inouye, and Pradeep Raviku- mar. On the (in)fidelity and sensitivity of explanations - NeurIPS 2019

---

> > ### Comment · Reviewer_Aokc · 2021-11-22
> > **Main concern is still the small sample size**
> >
> > Thank you for the detailed response.
> > While these clarifications make the manuscript significantly stronger, I still
> > think the empirical results are not very convincing unless the sample size is
> > closer to ~1000, or at least 300 for all the experiments.
> > Unless the sample size is not increased to this point, I do not feel
> > comfortable raising my recommended score any higher than 5.
> >
> > - Considering the runs over multiple seeds, could you report how many
> >   different runs (with the same samples) were conducted?
> > - As far as I understand, the pointing game in [1] only uses the highest value
> >   pixel. Maybe you could elaborate why you use the top 20 values. It would be
> >   interesting to see how these 3 metrics compare, although I do not think this
> >   is a *must*.
> > - Did you use VGG-16 and ResNet-18 for both post-hoc and ad-hoc experiments?
> >   Reading the manuscript, I had the impression that you only used one model
> >   each, but using both models for both setups would be a more satisfying
> >   coverage on the model side.

---

> > > ### Author Response · Authors · 2021-11-22
> > > **UPDATE: we have uploaded the rebuttal revision draft in which the sample sizes for the experiments were increased.**
> > >
> > > Thanks a lot for your reply. We have now uploaded the rebuttal revision draft. Given the timing and the computational power we had at hand, we could have increased the sample sizes as follows:
> > > - For post-hoc approaches:
> > >      - 1000 samples for the cascade randomization of model parameters.
> > >      - 60 samples for the class sensitivity experiment.
> > >      - 1000 samples for the Gini Index experiment.
> > >      - 100 samples for the Infidelity experiment.
> > >      - 200 samples for the explanation attacks against post-hoc approaches.
> > > - For ad-hoc approaches:
> > >      - 1000 samples for the cascade randomization of model parameters.
> > >      - 1000 samples for the Gini Index experiment.
> > >      - 1000 samples for the Infidelity experiment.
> > >      - 1024 samples for the explanation attacks against ad-hoc approaches.
> > >
> > > For the infidelity experiments for post-hoc approaches, we didn't manage to increase the sample size given the time we had. This experiment was done on the Imagenet dataset, and computing the Infidelity with square perturbation takes a lot of memory and also is computationally very heavy for high-dimensional data like Imagenet images. Similarly for the attacks against post-hoc approaches, since the attacks were against the explanation of Imagenet images, we could increase the sample size to 200. Concerning the class sensitivity experiment, for gathering the samples for this experiment, we had to take images from MS COCO dataset that contain 2 objects that are present among the Imagenet classes. Moreover, among those images, we prune the ones for which one or both of the objects are not among the top predicted classes by the network. Therefore, the process of finding such samples is quite time-consuming and we could only increase the sample size to 60.
> > >
> > > Here's also our comments on the bullet points mentioned in the reviewer's reply:
> > > - We used 5 different seeds.
> > > - The original pointing game [1] used only the pixel with the highest attribution. However, that experiment would be more appropriate to assess the localization of an explanation. Whereas here, we are interested in the ability of the explanations to discriminate between different classes present in the image. Therefore, we think that the top-20 values of the explanation would be a better indicator of the focus of the explanation (whether it aligns with the segmentation mask of the object for which the explanation was computed).
> > > - We used the VGG-16 and ResNet-18 for post-hoc and ad-hoc methods respectively. For the ad-hoc methods, we had the weights of the adversarially trained network and the CURE network for ResNet-18. That's why we used that network for evaluation. For the post-hoc methods, we experimented on ResNet-18 architecture for a small sample of Imagenet images and the results were similar to the VGG network.

---

> > > > ### Comment · Reviewer_Aokc · 2021-11-22
> > > > **Quality of the manuscript has increased**
> > > >
> > > > Thank you for your swift response and the updated manuscript.
> > > > While I think the sample size could still be higher for the VGG-16 infidelity
> > > > and explanation attack experiments, with the added changes I feel the empirical
> > > > results much more convincing.
> > > > Looking at the updated manuscript, I noticed a few things:
> > > >
> > > > - Table 3 is still missing the error values
> > > > - Table 5 reads "100" samples, but since you updated the experiment, it's
> > > >   probably just a minor oversight
> > > > - Unless I missed it, you should state that you averaged the results over 5
> > > >   runs with different seeds in the manuscript.
> > > > - I still think you should use 2 models for each post-hoc and ad-hoc
> > > >   experiments. To reduce runtime, you may use for example VGG-11.
> > > > - I understand finding suitable samples for the sensitivity experiment is
> > > >   challenging, but I think 20 was just too few. The results are much more
> > > >   convincing with the updated 60 samples.
> > > > - I may have also missed it, but the added error bars for the experiments
> > > >   should also be discussed in the manuscript.
> > > >
> > > > While not strictly necessary, I think adding your used hardware capacity and
> > > > runtime (to the supplement) may give a much better picture about scope of the
> > > > experiments to the reader.

---

> > > > > ### Author Response · Authors · 2021-11-23
> > > > > **We added the missing information in the manuscript**
> > > > >
> > > > > Thanks a lot for your detailed review and your feedback. We added the missing std values for the infidelity experiment on Imagenet and fixed the typo in the caption of Table 5. For the Infidelity experiment on Imagenet, the standard deviation values are quite high. We believe that by using more samples for this experiment, it would be possible to reach more stable values for the Infidelity. However, still, the average values point us to the fact that the post-hoc smoothing methods considered in this work help to improve the infidelity of explanation. For the sake of completeness, we also included the results of the Infidelity test for the explanations of a VGG-16 network trained on CIFAR-10 in the appendix (Table 8). For the samples from CIFAR-10, we can observe a much more significant improvement of Infidelity caused by post-hoc smoothing approaches.
> > > > >
> > > > > Thanks again for your suggestions and feedbacks. We hope that your concerns have been addressed and you would reconsider your evaluation.

---

### Official Review · Reviewer_Sfxr · 2021-11-02

**Correctness:** 3
**Technical Novelty And Significance:** 2
**Empirical Novelty And Significance:** 2
**Recommendation:** 5
**Confidence:** 4

**Main Review:**

### Strength
This paper presents sanity check, non-Lp adversarial attacks, and sparsity measurement on SmoothGrad, Uniform Gradient and $\beta$-smoothing. The results are generally interesting to the community.
Using the Gini Index to measure the sparsity is an interesting idea that might be useful for the follow-up works. However, I have several questions regarding this metric. I will elaborate in the next subsection.

### Weakness
- It is not surprising that Lp-based defenses (smoothing) are not robust to non-Lp threat models in the community. The authors aim to present evidence that smoothing techniques are not robust enough because the proposed non-Lp attacks can break the explanations, which by itself seems to be an unfair comparison for the prior work. In fact, the contributions showing the proposed non-Lp attacks can break Lp defense is not a new observation.

- The proposed attacks do not seem to produce very different results. In Fig 5, resulting adversarial attacks from new technologies remain very similar to Delta attack. In Fig 4, it seems that the new attacks only make more than 0.05 difference with the Delta attack on $\beta$-smoothing in cosine distance. On other attributions, the improvement seems to be minimal.

- Gini Index for evaluating sparseness. The motivation to use Gini Index seems to be fair, however, a lot of prior works have proposed several different metrics [1, 2, 3, 4] in measuring the sparseness (or the concentrations and the localizations on the relevant features) of attributions. At least some discussions and justifications of the proposed metrics should be included. The robustness-related and fidelity-related evaluations are motivated from the paper’s main idea, understanding if smoothing techniques provide faithful explanations, however, the transition to study sparsity is somewhat sudden to me and it seems to be unconnected from the previous content, only because the following reason authors provide: “To create explanations that are human-accessible, it is advantageous to have a sparse explanation map”

- Some related prior work, i.e. ROAR [5] , that finds smoothing techniques do not create significant degeneration to explanations should be included and discussed, especially when the paper is trying to provide the shortcoming of the smoothing techniques.


[1] A. Chattopadhay, A. Sarkar, P. Howlader and V. N. Balasubramanian, "Grad-CAM++: Generalized Gradient-Based Visual Explanations for Deep Convolutional Networks," 2018 IEEE Winter Conference on Applications of Computer Vision (WACV), 2018, pp. 839-847, doi: 10.1109/WACV.2018.00097.

[2] Poppi, S., Cornia, M., Baraldi, L., & Cucchiara, R. (2021). Revisiting The Evaluation of Class Activation Mapping for Explainability: A Novel Metric and Experimental Analysis. 2021 IEEE/CVF Conference on Computer Vision and Pattern Recognition Workshops (CVPRW), 2299-2304.

[3] Wang, H., Wang, Z., Du, M., Yang, F., Zhang, Z., Ding, S., Mardziel, P., & Hu, X. (2020). Score-CAM: Score-Weighted Visual Explanations for Convolutional Neural Networks. 2020 IEEE/CVF Conference on Computer Vision and Pattern Recognition Workshops (CVPRW), 111-119.

[4] Fong, R., Patrick, M., & Vedaldi, A. (2019). Understanding Deep Networks via Extremal Perturbations and Smooth Masks. 2019 IEEE/CVF International Conference on Computer Vision (ICCV), 2950-2958.

[5] Hooker, Sara et al. “A Benchmark for Interpretability Methods in Deep Neural Networks.” NeurIPS(2019).


**Summary Of The Paper:**

The main contributions of this paper are a series of empirical results on smoothed attribution methods that are designed to show several smoothing techniques in the previous literature may produce worse explanations and these smoothing techniques are also not robust to non-Lp attacks.


**Summary Of The Review:**

In summary this paper presents a series of interesting empirical results for smoothing techniques. However, I find the empirical results for the goodness of the smoothed attributions are not convincing to me, and the conclusions from the non-Lp attacks are not very new to the community, I remain negative for this paper.

---

> ### Author Response · Authors · 2021-11-17
> **The post-hoc methods were not explicitly designed to be robust against additive attacks. Therefore, the fact that non-additive attacks can weaken the robustness of post-hoc methods is indeed a new observation.**
>
> We thank the reviewer for their thorough feedback. Here's our response to the points raised by the reviewer under the weakness section:
> - It has been shown in the context of prediction robustness that models that have been adversarially trained on $\ell_p$ perturbed examples are less robust against non-$\ell_p$ threat models like spatial transformation (StAdv). It has been shown that combining additive and non-additive threat models can create even stronger attacks that can reduce the accuracy of $\ell_p$ adversarially trained networks even more (e.g, Table 2 in https://arxiv.org/pdf/1902.08265.pdf and Table 1 in https://arxiv.org/pdf/1906.00001.pdf). However, it is not completely trivial that these observations can also be transferred to the case of the robustness of explanations. A line of work in the explainability community has already shown that adversarially trained networks or networks that are trained to regularize the spectral norm of the hessian (e.g, CURE and SSR) provide robust explanations. These works only showed robustness against explanation attacks based on $\ell_p$ threat models. To the best of our knowledge, this is the first work that explored the efficacy of explanation attacks based on non-$\ell_p$ threat models against the explanation of such networks. Moreover, robust explanation methods like $\beta$-smoothing or smooth gradient were not explicitly designed to be robust against $\ell_p$ based explanation attacks. Therefore, the fact that explanation attacks based on combining additive and non-additive threat models can weaken their robustness is indeed a new observation.
>
> - For the $\beta$-smoothing method, the new non-additive attacks can actually create a 0.1 difference in the cosine distance (cosd) value compared to the additive attack. By comparing the results of the attacks against the gradient and the $\beta$-smoothing methods (Figure 4 left), we can observe that using the non-additive attacks against the $\beta$-smoothing method, we can actually reduce the cosd value to the same level that was achieved by attacking the gradient method (which is not robust). In other words, attacking the $\beta$-smoothing method with the combination of additive and non-additive attacks makes it as non-robust as the gradient method. We achieve this result while we are not losing the original and adversarial image similarity, compared to when attacking the $\beta$-smoothing method with the additive attack (Figure 4 right). For the case of the smooth and uniform gradient methods, the additive attack can already reduce their cosd value to the same level as the gradient method. Therefore, for these methods, the difference that the non-additive attacks can make is less significant but they still can reduce the cosd value as compared to the additive attack (Figure 4 left, second and third column).
>
> - In this work, we wanted to assess different quality aspects of smoothed explanations using the different quality metrics available in the literature. In that sense, the explanation sparseness is a quality metric that has been claimed to be desirable for the explanations in the prior works [1, 2]. In this regard, we brought in the sparsity measure in order to create a more complete picture of different quality aspects of explanations and to observe how the explanation methods we investigated in this work perform compared to each other in terms of these quality measures. In particular, we chose to use the Gini Index to measure the sparseness of explanations, as it is a general measure that can be applied to the output of any explanation method. We didn't use the pointing game in this part, as we were more interested in how sparse (and hence concise) the explanations are rather than how well they can localize the object corresponding to the target class.
>
> - Indeed the ROAR paper introduces an interesting metric that is more fair compared to the traditional benchmarks for interpretability methods that remove features with high attributions and observe how the output of the network degrades. However, the ROAR experiment is very expensive to perform and the results of the paper actually show that classic smoothing methods like smooth-grad and uniform-grad (which were investigated in our work) do not perform better compared to the vanilla gradient method (in terms of the ROAR evaluation). Moreover, we would like to emphasize that in this work we are not evaluating all smoothing methods for explanations in general, and our results and conclusions do not necessarily extend to all of the smoothed explanation methods.
>
>
> [1] Prasad Chalasani, Jiefeng Chen, Amrita Roy Chowdhury, Xi Wu, and Somesh Jha. Concise explanations of neural networks using adversarial training. In Proceedings of the 37th International Conference on Machine Learning, ICML 2020.
>
> [2] Christoph Molnar. Interpretable Machine Learning. 2019.

---

> > ### Author Response · Authors · 2021-11-26
> > **Could you please share with us whether your concerns have been addressed or not?**
> >
> > Dear reviewer,
> >
> > Once again, thanks for your detailed feedback. In our reply to your previous comments, we tried to address your questions and concerns. We would very much appreciate it if you could share with us your opinion about our reply. Did our reply address your questions and concerns?
> >
> > Thanks a lot.

---

> > ### Comment · Reviewer_Sfxr · 2021-11-26
> > **Thanks for the response and the revision**
> >
> > Dear authors
> >
> > I am sorry for my late reply. I appreciate the revision and the thoughtful response. I decide to maintain my score after reading the authors' response but I appreciate that the authors have helped to understand the paper better.
> >
> > 1. I believe that the $\ell_p$-threat model and their corresponding defense has been shown to be not robust to non-$\ell_p$ threat model, which does not seems to be a significant contribution especially the target explanations are all gradient-based, as most of the attacks are using gradient-based iterations as well. It is a good observation for gradient-based attributions, but that is hard to claim as a weakness or criticism for smoothing techniques used in attributions because these smoothing techniques indeed can defend the $\ell_p$ threat model that motivates them.
> >
> > 2. I agree with the authors that the new attack seems to work better for $\beta$-smoothing but on the other hand, the improvement on other defenses seem to be not that obvious. Therefore, I think the authors may want to slightly change the tone in the contribution to narrow the effectiveness of the new attack on all attributions to the $\beta$-smoothing. Though i am more curious why $\beta$-smoothing is worse from some theoretical perspective.
> >
> > 3. For the Gini index discussion: maybe this is just my read, but isn't spareness in visual explanations usually referred to (and motivating) the alignment between the attribution map and the relevant objects in the images? I am a bit unclear why would people, in the other way around, want an attribution to be sparse but not concentrates on the relevant objects.

---

> > > ### Author Response · Authors · 2021-11-29
> > > **The papers that introduced these smoothing methods did not claim that they are only robust to $\ell_p$ threat models.**
> > >
> > > Dear reviewer,
> > >
> > > Thanks for your reply.
> > >
> > > The smoothing methods we investigated in this work, did not have an explicit motivation to be only robust against explanation attacks based on additive models. $\beta$-smoothing [2] method performs a global smoothing and smooth gradient, and uniform gradient [1] perform a local smoothing on the predictor function which is not necessarily related to the $\ell_p$ threat models. Similarly, SSR regularises the maximum absolute eigenvalue of the hessian which is again not necessarily related to the $\ell_p$ threat models. Therefore, this is not obvious that these methods are only robust against $\ell_p$ threat models. Specifically the post-hoc methods do not even have an equivalent robustness method in the context of adversarial examples. So we cannot easily extend the conclusions from the adversarial robustness context to the explanation robustness for the post-hoc methods considered in this paper. Moreover, the papers that introduced these smoothing methods only tested these methods against the $\ell_p$ threat model but did not claim that they are only robust to $\ell_p$ threat models. Therefore the question of whether these methods are also robust to non-$\ell_p$ threat models or not remains unanswered.
> > >
> > > About the Gini index discussion: The alignment between the explanation map and the relevant object in the image is also referred to as the “localization” ability of the explanation. The localization ability and sparseness are two different and desired properties for explanations. For Evaluating the sparseness, the Gini index has been used in the literature [4] and for evaluating the localization ability the pointing game [5] experiment is usually used. It might be possible to come up with a better evaluation metric that can measure both of these properties (localization and sparseness) simultaneously. However, in this work, we used the evaluation metrics already introduced in the literature.
> > >
> > > [1] Zifan  Wang,  Haofan  Wang,  Shakul  Ramkumar,  Piotr  Mardziel,  Matt  Fredrikson,  and  Anupam  Datta. Smoothed geometry for robust attribution. Advances in Neural Information Processing Systems  33, NeurIPS  2020, December 6-12,  2020
> > >
> > > [2] Ann-Kathrin  Dombrowski,  Maximilian  Alber,  Christopher  J.  Anders,  Marcel  Ackermann,  Klaus-RobertM ̈uller,  and  Pan  Kessel.    Explanations can be manipulated and geometry is to blame. Advances  in  Neural  Information  Processing  Systems  32, NeurIPS  2019
> > >
> > > [3] Prasad  Chalasani,  Jiefeng  Chen,  Amrita  Roy  Chowdhury,  Xi  Wu,  and  Somesh  Jha.   Concise explanations of neural networks using adversarial training.   InProceedings  of  the  37th  International  Conference  on Machine Learning, ICML 2020
> > >
> > > [4] Richard Zhang, Phillip Isola, Alexei A. Efros, Eli Shechtman, and Oliver Wang. The unreasonable effectiveness of deep features as a perceptual metric. In2018 IEEE Conference on Computer Vision and Pattern Recognition, CVPR 2018

---

### Official Review · Reviewer_dEQe · 2021-11-03

**Correctness:** 4
**Technical Novelty And Significance:** 3
**Empirical Novelty And Significance:** 3
**Recommendation:** 6
**Confidence:** 4

**Main Review:**

The paper touches on a very important problem - the quality and understanding of (smoothed) explanations. It is very well written and an easy to follow.

It has many positives. I particularly liked the approaches to assess robustness as they seems very reasonable and widely applicable, for example using LPIPS and perception and a way of giving an interesting new perspective. The main to areas for improvement / clarification are as follows.

- Generalization. How do the results extend beyond the specific networks chosen? It is difficult to understand how relevant the conclusions are in general and whether the example configuration in influencing the conclusions too much.

- Formalization. For me, the paper lack in the formalization of the problem. After reading the paper I don't fully have a clear notion of what makes an explanation of this form high quality and the exact properties that one should be looking for to assess it. Of course, the metrics reported are proxies to it, but the actual objetive is not clear to me.



**Summary Of The Paper:**

The paper presents experiments for both post-hoc and ad-hoc explainers to better understand their quality and robustness.

**Summary Of The Review:**

Good paper but I expected a bit more in terms of formalizing quality.

---

> ### Author Response · Authors · 2021-11-17
> **Formalising the explanation quality as a whole is a challenging task.**
>
> We thank the reviewer for their comments. Here's our comment on the two main points raised by the reviewer:
> - Generalization: In the experiments, we used two network architectures, namely, VGG-16 and Resnet-18. These networks have a similar structure to other CNN and Resnet networks, so we believe that the results are representative and can be extended to other such architectures as well. We chose these two networks as they have been widely used in the context of image classification and have shown good performance on benchmark datasets such as Imagnet and CIFAR-10. It is worth mentioning that transformer networks have recently also been explored in the context of image classification and have been shown to attain comparable results compared to the state-of-the-art convolutional networks (e.g, https://arxiv.org/pdf/2010.11929.pdf). Moreover, recent works proposed new explainability methods based on transformer networks (e.g, https://arxiv.org/pdf/2012.09838.pdf) and demonstrated that they have a clear advantage over the existing explainability methods. Exploring such networks and explainability methods was beyond the scope of this work, however, we believe that it is an interesting direction for future work.
>
> - Formalizing the explanation quality: Assessing the quality of the explanations is a challenging task in general. Different works in the explainability literature have argued about qualities desirable for an explanation and defined quality tests and sanity checks to assess those qualities. Each of these quality tests only evaluates a specific property of an explanation. We also tried to use a range of quality tests and metrics to be able to compare different aspects of the explanation methods we considered in this work. Formalizing the quality of explanations as a whole is challenging and we should be specific about which aspect of the quality we are talking about.

---

### Decision · Program_Chairs · 2022-01-20

**Decision:**

Reject

**Comment:**

The paper conducts a series of empirical studies to evaluate the robustness of smoothed attribution methods. Although the reviewers think this is an important direction, there are several concerns about the experimental settings, such as the sample size and the models to be tested. Also, one of the main finding that Lp based smoothing methods are non-robust to non-Lp norm perturbations is well known and is not that surprising.